# Role and Treatment of Insulin Resistance in Patients with Chronic Kidney Disease: A Review

**DOI:** 10.3390/nu13124349

**Published:** 2021-12-02

**Authors:** Akio Nakashima, Kazuhiko Kato, Ichiro Ohkido, Takashi Yokoo

**Affiliations:** Division of Nephrology and Hypertension, Department of Internal Medicine, The Jikei University School of Medicine, Tokyo 105-8461, Japan; kazu.j429@gmail.com (K.K.); iohkido@jikei.ac.jp (I.O.); tyokoo@jikei.ac.jp (T.Y.)

**Keywords:** chronic kidney disease, insulin resistance, vitamin D, cardiovascular disease

## Abstract

Patients with chronic kidney disease (CKD) and dialysis have higher mortality than those without, and cardiovascular disease (CVD) is the main cause of death. As CVD is caused by several mechanisms, insulin resistance plays an important role in CVD. This review summarizes the importance and mechanism of insulin resistance in CKD and discusses the current evidence regarding insulin resistance in patients with CKD and dialysis. Insulin resistance has been reported to influence endothelial dysfunction, plaque formation, hypertension, and dyslipidemia. A recent study also reported an association between insulin resistance and cognitive dysfunction, non-alcoholic fatty liver disease, polycystic ovary syndrome, and malignancy. Insulin resistance increases as renal function decrease in patients with CKD and dialysis. Several mechanisms increase insulin resistance in patients with CKD, such as chronic inflammation, oxidative stress, obesity, and mineral bone disorder. There is the possibility that insulin resistance is the potential future target of treatment in patients with CKD.

## 1. Introduction

Cardiovascular disease (CVD) is the main cause of death among patients with diabetes mellitus and chronic kidney disease (CKD). One of the main causes of CVD is insulin resistance (IR), which is a balance between calorie intake and consumption. Although IR is affected by genetic predisposition, lifestyle factors such as dietary habits and daily exercise affect IR through hypertrophy of fat cells, ultrafiltration of inflammatory cells into adipose tissue, and inappropriate secretion of adipokines [1]. Patients with a higher IR status have higher mortality and morbidity rates than those with a lower IR status [2,3]. Patients with CKD have a higher IR status than patients without because of chronic inflammation, uremic toxins, and vitamin D deficiency [4]. In addition, a recent study reported that CKD-mineral bone disorder (MBD), phosphorus, and fibroblast growth factor-23 (FGF-23) also affect IR [5]. There is the possibility that IR is improved by CKD-MBD, such as phosphate binder and vitamin D. Vitamin D has also received a great deal of attention in recent years in IR. Clinical studies have been conducted to improve IR with activated vitamin D analog and vitamin D supplementation. Although the mechanism of IR has been revealed in recent years, the detailed mechanism is unknown. Therefore, this review summarizes the importance and mechanism of IR in CVD and discusses the current evidence regarding IR and patients with CKD.

### 1.1. Mechanisms of IR

Insulin mainly targets the liver, skeletal muscle, and fat cells and regulates glucose metabolism via the insulin receptor. Insulin receptors are distributed in various tissues and play important physiological roles. IR is defined as the inability of cells to respond to the action of insulin. As patients with IR have less adipose tissue insulin sensitivity, skeletal muscle, and liver, the blood glucose is elevated due to IR. The main causes of IR are obesity and overnutrition, which induce imbalance of adipocytokines, lipotoxicity, and chronic inflammation.

### 1.2. Adipocytokines and IR

Several factors increase IR. Adipocytes not only contain triglycerides but also secrete several adipocytokines. In obese patients, the accumulation of triglycerides induces hypertrophy of adipocytes. This hypertrophy affects the secretion of adipocytokines and causes IR. Tumor necrosis factor-alpha (TNF-α), free fatty acids (FFAs), and leptin may increase IR. TNF-α is an adipocytokine secreted by adipocytes, and its concentration increased in obese animal models or patients with obesity. The secretion of TNF-α inhibits insulin activity by inducing phosphorylation of insulin receptor substrate-1 (IRS-1) [6,7].

Additionally, the administration of soluble TNF-α receptors attenuated IR in fatty model animals [6]. Although skeletal muscle and the liver are the main organs where IR develops, FFAs are essential sources of lipids, and they play an important role in IR. Adipocyte atrophy induces lipolysis and releases FFA [8]. FFAs are thought to accumulate as skeletal muscle intracellular (intramyocellular lipid) and intrahepatic lipids and induce IR. Leptin is an adipocytokine that regulates appetite and induces the consumption of energy, increasing insulin action. A previous study reported the effect of leptin on obesity and decreased IR in animal models of diabetes. In addition, leptin administration decreased IR in a mouse model. A clinical study also reported a relationship between IR and serum leptin levels. However, the administration of leptin in obese patients was not beneficial.

### 1.3. Lipotoxicity and IR

Lipotoxicity, defined by Unger et al., is a condition in which free FFAs suppress insulin secretion of glucose responsivity [9]. FFAs have been shown to inhibit insulin action in insulin-targeted organs, such as the liver, skeletal muscle, and adipose tissue [10]. When a fat emulsion is intravenously administered to healthy patients to increase the blood FFA concentration, insulin’s glucose uptake in peripheral tissues is suppressed, resulting in worse IR [11]. FFAs taken up into cells are converted to diacylglycerol (DAG) and activate protein kinase C (PKC) and c-Jun N-terminal kinase (JNK). PKC and JNK induce IR by phosphorylating the serine residue of IRS-1 and suppressing insulin signal transduction.

Excess FAA uptake in the liver induces ectopic fatty accumulation and results in non-alcoholic fatty liver disease (NAFLD) [12]. In patients with NAFLD, the reduction of insulin receptor signaling through DAG and PKC accumulate forkhead transcription factor (FOXO) 1 and FOXO2 and upregulates gluconeogenesis [13].

Chronic inflammation and other mechanisms of IR:

In patients with obesity, inflammatory cytokines, such as interleukin (IL)-1, IL-6, and TNF-α, increase and are reported to induce higher IR conditions [14,15]. Various inflammatory cells, including macrophages, infiltrate the adipose tissue of obese people; many inflammatory cytokines suppress insulin activity, resulting in increased IR [16,17].

A recent clinical study showed that hepatitis C virus clearance by direct-acting antivirals improved IR [18,19]. In addition, another study reported that hepatitis C treatments reduce CVD events in the prediabetic population [20]. These results suggest the importance of hepatitis C virus for glycemic conditions and the possibility as a future therapeutic target for diabetes mellitus.

## 2. Assessment of IR in Patients with Diabetes Mellitus

Clinicians usually use glycated hemoglobin A1c and glycated albumin to confirm patients’ glucose levels and the therapeutic effects of treatment. However, these analyses cannot be used to assess IR. The homeostasis model assessment of insulin resistance (HOMA-IR) and serum insulin levels is usually used to assess the condition of insulin resistance. HOMA-IR is calculated from the fasting blood insulin level in the early morning and the fasting blood glucose level and is often used clinically as a simple index of IR. It correlates well with IR determined by the clamp method when the fasting blood glucose level is 140 mg/dL or less. The normal value of HOMA-IR is less than 1.6, and patients with a value higher than 2.5 are regarded as having a high IR status. HOMA-IR is a method in which only one blood sample is required, and the cost of the analysis is low. However, HOMA-IR may not be useful in patients receiving basal insulin subcutaneous therapy because the actual status of IR is not reflected. Fasting IR is normal at 2–10 μg/dL, and if it exceeds that, it is considered to be compensated for by insulin hypersecretion. In type 2 diabetes and its reserves, there is often a delay in additional insulin secretion. Even if the insulin index decreases 30 min after loading, insulin secretion is high after 60 or 120 min. This condition often becomes a measure of IR.

The glucose clamp method was devised by DeFronzo et al. and is regarded as the gold standard method for assessing IR [21]. However, the technique is complicated because it requires time for equipment and inspection, making it difficult to use in daily medical treatment and large-scale clinical studies. With the patient in a fasting and resting supine position, insulin is infused from one arm, and the blood insulin concentration is maintained at 100 μU/mL. At the same time, glucose is administered (usually 10% glucose is used), blood is collected from the opposite arm, and the blood glucose level is maintained at 100 mg/dL 90–120 min later. The glucose administration rate (glucose infusion rate, mg/kg/min) is regarded as the glucose absorption rate in response to insulin stimulation. This rate is usually low in patients with IR.

## 3. IR in Patients with CKD

Previous studies have reported that IR is higher in patients with CKD than in the general population. IR has already been reported to occur during stage 1 of CKD [22], although another large-scale study (*n* = 17,157) reported that a decreased estimated glomerular filtration rate (eGFR) was not related to increased IR [23]. Another study reported a significant association between IR and eGFR, but this association disappeared after adjusting for variables in the multivariate analysis, including the body mass index (BMI) [24]. As the above-mentioned studies included patients with early stage CKD, the association between IR and kidney function was affected by BMI in patients with normal kidney function (Figure 1).

In patients with CKD stages 3–4, HOMA-IR was high, and the primary determinant of IR in this population was BMI, specifically fat mass [25]. Among patients undergoing dialysis, HOMA-IR was also high and correlated with inflammatory markers [26]. CKD progression and IR have also been analyzed in several studies. HOMA-IR and metabolic syndrome are risk factors for CKD progression and reduced eGFR [27]. One study reported that a high level of HOMA-IR was associated with the development of albuminuria in relatively healthy individuals without diabetes mellitus [28]. Another study also showed that the urinary albumin-creatinine ratio is associated with HOMA-IR after adjustment for kidney function, BMI, blood pressure, and diabetes mellitus [29]. A recent interventional study showed that multifactorial treatment, not only glycol-metabolic control but also hypertension and dyslipidemia by non-pharmacological and pharmacological approaches, prevent CVD events in diabetic CKD patients. This study suggests the importance of multifactorial approach in treating CKD patients [30].

In a study using renal tissue, IR was reported to correlate with renal tubular interstitial fibrosis and arteriolosclerosis of renal blood vessels among renal tissue disorders [31]. The increase in adiponectin, an inflammatory adipocytokine, such as TNFα, IL-1β, and IL-6, has a large effect on the renal disorders associated with obesity, causing an increase in the mesangial matrix and inflammation of the renal tubular epithelium [32]. The IRS-PI3K AKT, downstream of insulin binding to its receptor, is believed to cause IR in CKD [33,34].

Uremic toxins accumulate in the body as renal function declines, but their origins vary from endogenous metabolites, gut microbiota metabolites, and exogenous metabolites. Recently, uremic toxins such as blood urea nitrogen (BUN), *p*-cresyl sulfate, indoxyl sulfate, and asymmetric dimethylarginine (ADMA) have attracted attention as causes of renal IR. BUN affects reactive oxygen species (ROS), and tyrosine of the IRS causes ROS production, which has been shown to suppress phosphorylation and PKB/Akt phosphorylation and attenuate glucose transport in CKD model mice [35].

There are several theories for kidney dysfunction and IR upregulation. First, uremic toxins increase IR through oxidative stress and ROS activity [35]. In patients with CKD, the number of insulin receptors, insulin coupling ability, and tyrosine kinase activity are important factors of IR. Second, uremic toxins induce ROS production, cause IR, and increase adipokines.

ADMA also increases renal failure and correlates with the degree of coronary artery calcification, an arteriosclerotic disease [36]. However, when corrected by HOMA-IR, the correlation disappears, so the involvement of ADMA in coronary artery calcification with IR is associated with the upstream of ADMA. ADMA is an antagonist of endogenous NO. It is decomposed by dimethylarginine dimethylaminohydrolase (DDAH), and DDAH has been reported in the ADMA system involved in stimulating insulin secretion in pancreatic β cells [37].

A previous study reported that ADMA treatment led to oxidative stress and steatosis, whereas overexpression of DDAH decreased palmitic acid-induced steatosis, oxidative stress, and inflammation. These results suggest that the ADMA/DDAH pathway is important for IR and NAFLD and that DDAH is a potential therapy for IR [38].

Insulin-induced glucose in adipocytes is associated with decreased DDAH levels and a concomitant increase in ADMA levels. It suppresses metabolism, but these changes are modified by administering the aldosterone antagonist spironolactone [39].

In patients with primary aldosteronism, after treatment with adrenalectomy or a mineralocorticoid receptor antagonist for primary aldosteronism, the insulin response to glucose increases, and insulin clearance decreases. As this effect is independent of kidney function and cortisol levels, a previous study suggested that aldosterone affects insulin secretion and IR [40].

## 4. CKD-MBD and IR

CKD-MBD is a concept that involves abnormalities in phosphorus, calcium, the parathyroid hormone (PTH), and other bone markers and aims to prevent CVD events and fractures [41]. The main pathological mechanism of CKD-MBD is the elimination of phosphorus in the renal tubules and the activation disorder of vitamin D [42]. The association between CKD-MBD and IR has also been investigated in recent years (Figure 2).

### 4.1. Vitamin D and IR

Vitamin D is a fat-soluble steroid hormone that promotes dietary calcium and phosphorus absorption through the vitamin D receptor (VDR) [43]. VDR is expressed in whole- body organs, including the heart, liver, kidney, blood vessels, and central nervous system. Moreover, the beta cells in the pancreas also have the VDR and play an important role [44]. Several mechanisms of the association between vitamin D and IR have been reported. Vitamin D stimulates insulin receptors and is thought to be involved in glucose transport by increasing the reactivity to insulin [45]. As pancreatic beta cells are affected by cytokine-induced apoptosis, a high inflammation status causes worsening glycemic control. Vitamin D can decrease the effects of systemic chronic inflammation and protect against beta-cell cytokine-induced apoptosis by directly modulating the expression and activity of cytokines [46,47]. Another study reported that VDR activation by a vitamin D analog reduced liver inflammation and improved IR [48]. Additionally, vitamin D improves hyperactivity of the aldosterone effect through the renin-angiotensin system, activates the function of beta cells in the pancreas, and improves IR [49,50]. Low vitamin D levels predicted future glycemia and IR in a 10-year follow-up cohort study [51]. Other studies have shown that low vitamin D levels are associated with IR [52,53,54].

### 4.2. Phosphorus, FGF-23, and IR

FGF-23, an endocrine hormone secreted by osteocytes, regulates vitamin D metabolism and phosphorus balance through renal tubules. High levels of FGF-23 are associated with all-cause mortality and CVD events in patients with CKD, CKD, and dialysis [55,56,57]. The association between FGF-23 and IR has been studied recently. Several studies have reported that FGF-23 contributes to IR in patients with CKD and obesity [58,59,60,61]. Resistin, expressed in monocytes and macrophages, is the key peptide of IR and is involved in inflammatory processes [62]. We have reported that resistin is associated with FGF-23 in patients with type 2 diabetes mellitus [63]. Because this association is significant after adjustment for 25-hydroxyvitamin D (25OHD), C-peptide, ghrelin, leptin, PTH, FGF-23, and resistin are related to each other. Although the detailed mechanisms of FGF-23 and resistin have not been elucidated, it is considered that chronic inflammation of FGF-23 may induce a higher resistin status. Although phosphorus restriction and phosphate binder decrease serum FGF23 levels, no study investigated whether phosphorus and FGF23 control improved resistin levels in IR status.

## 5. Treatment of IR

There are two main approaches to the treatment of IR: weight loss and pharmacological therapy. Inflammatory cytokines are produced in obese adipose tissue, and infiltration of immune cells is observed; this causes abnormal glucose metabolism and IR. In addition, the gut microbiota of patients with obesity differs compared to that in the general population. This alteration causes intestinal barrier dysfunction, secretory deficiency of incretin in the intestine, and activation of chronic inflammation. These changes worsen the conditions of IR and cause NAFLD and diabetes mellitus [64,65,66]. Short-chain fatty acids produced by the breakdown of dietary fiber caused by intestinal bacteria promote energy consumption and increase appetite-suppressing hormones [67].

For severe obesity with a BMI of 35 kg/m^2^ or higher, the goal is to reduce weight by 5–10% with a calorie intake of 20–25 kcal or less per standard daily body weight. As exercise is an important method to reduce body weight, several guidelines recommend regular exercise. To reduce 5–7.5 kg weight, the American College of Sports Medicine recommends 225–420 min of aerobic exercise per week [68].

Thiazolidine improves IR by inhibiting the peroxisome proliferator-activated receptor (PPAR) that regulates fatty cells. It was revealed that thiazolidine increases the blood adiponectin concentration while reducing the expression of inflammatory cytokines, such as TNF-α and monocyte chemoattractant protein-1, and improving IR [69,70]. Pioglitazone was found to reduce CVD events by 10%. Furthermore, the combined incidence of all-cause mortality, non-fatal myocardial infarction, and stroke, which are hard endpoints, was reduced by 16% [71]. When pairing type 2 diabetic patients with and without CVD, the amount of change in the carotid intima-media complex thickening (mean IMT) in thiazolidine was verified. The mean IMT of the glimepiride group increased by 0.012 mm, whereas that of the pioglitazone group decreased by 0.001 mm. It was revealed that pioglitazone suppressed the progression of IMT [72]. However, since there has been concern about using thiazolidine for fracture, bladder cancer, and fluid retention, it should be used carefully [70].

Sodium-glucose transport protein 2 (SGLT2) inhibitors suppress glucose reabsorption in the proximal tubule and improve blood glucose levels by increasing glucose excretion in the urine [73]. SGLT2 decreases weight as a result of catabolic fat burning and improves IR [74]. In an animal diabetic model study, SGLT2 improved IR by stimulating muscle glucose uptake [75].

### Is CKD-MBD the Potential Option for Treating IR?

As aforementioned, CKD-MBD, including vitamin and FGF-23, is closely associated with IR. There are two types of research on vitamin D for IR: activated vitamin D analog and vitamin D supplementation. Spoto et al. reported whether paricalcitol improves HOMA-IR in CKD stage 3–4 patients. Although paricalcitol decreases serum parathyroid hormone levels, there are no significant differences in IR between paricalcitol and placebo [76]. Mahmoudi et al. reported that calcitriol, an activated vitamin D analog, improved HOMA-IR in patients with NAFLD [77]. Another study about activated vitamin D analog for IR reported positive effects in NAFLD patients [78]. However, a 2-month crossover randomized controlled trial about paricalcitol or placebo did not show positive effects for IR in the vitamin D group [79]. Although interventional studies about activated vitamin D analogs are relatively fewer than vitamin D supplementation, these studies almost showed that the activated vitamin D analog improved IR.

The studies about vitamin D supplementation, cholecalciferol or ergocalciferol, are much more than activated vitamin D analogs and have shown several results. One study reported that vitamin D supplementation, cholecalciferol, improved HOMA-IR and the insulin sensitivity check index in premenopausal women [78]. On the other hand, there are opposite results for vitamin D regarding IR [48,80,81]. Several meta-analyses of vitamin D supplementation for IR have been reported. He et al. reported that although vitamin D supplementation had no significant effect on IR, insulin resistance was improved for those with 25(OH)D ≥30 ng/mL by stratified analysis [82]. Other studies also suggested vitamin D supplementation improves IR in prediabetics and polycystic ovary syndrome [83,84]. The evidence of activated vitamin D analogue and vitamin D supplementation for IR was insufficient, and well-designed clinical studies are needed.

Although phosphorus and FGF-23 are associated with IR, these results have been investigated in mainly observational studies [59,61,63,85]. New evidence is needed to determine whether diet phosphate control or phosphate binders improve IR. Phosphate binders are divided into calcium-containing phosphate binders and non-calcium-containing phosphate binders, which have different effects on vascular calcification [86,87]. As earlier studies reported, calcium-containing phosphate and non-calcium-containing phosphate binders decrease FGF23 [88,89,90]. Future studies that compare phosphate binder and placebo for IR are needed. In addition, researchers should investigate the differences between non-calcium-containing phosphate and calcium-containing phosphate binders for IR.

This review article has some limitations. First, the clinical evidence of IR is lacking, especially randomized controlled trials that explored the possibility of decreasing IR. Therefore, future well-designed studies will be needed. Second, since IR is regarded as a risk factor for CVD events, few data exist on other clinical important events, such as fracture and malignancy.

## 6. Conclusions

IR is important not only for diabetes mellitus but also for CKD. CKD-MBD is associated with IR and has the potential to be a treatment option. Future well-designed studies are required to investigate this further.

## Figures and Tables

**Figure 1 nutrients-13-04349-f001:**
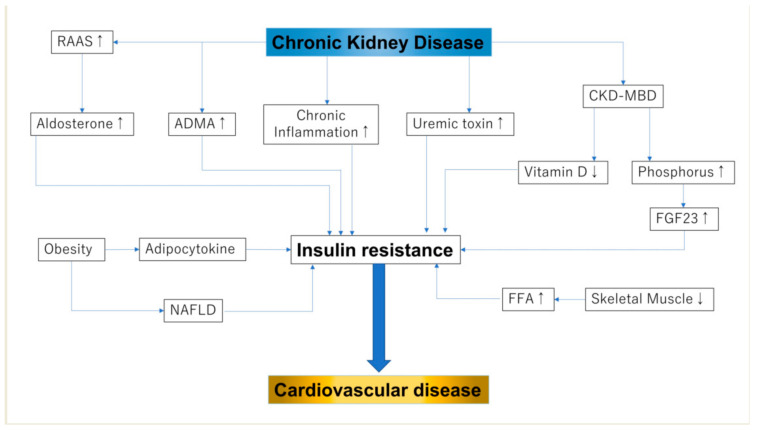
Insulin resistance and chronic kidney disease: RAAS, renin–angiotensin–aldosterone sys; FGF-23, fibroblast growth factor 23; NAFLD, non-alcoholic fatty liver disease; FFA, free fatty acid. In this figure, up-arrow shows enhanced activity and down-arrow shows reduced activity.

**Figure 2 nutrients-13-04349-f002:**
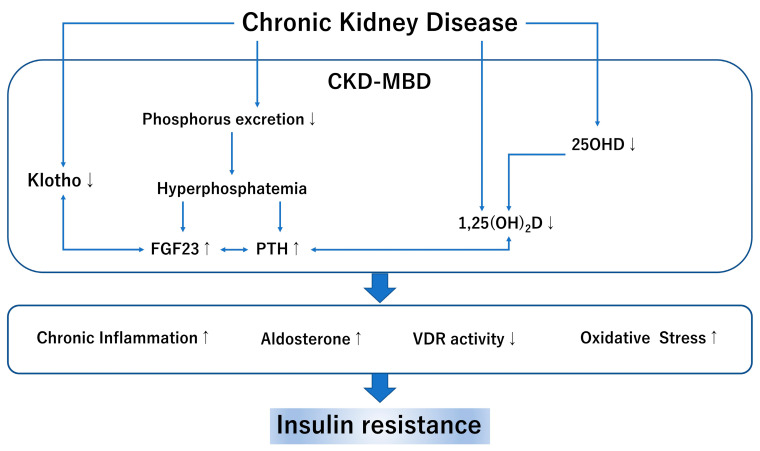
CKD-MBD and insulin resistance: CKD, chronic kidney disease; MBD, mineral bone disorder; FGF23 fibroblast growth factor 23; PTH, parathyroid hormone; 25OHD, 25 hydroxyvitamin D; 1,25(OH)_2_D, 1,25 dihydroxy vitamin D; VDR, vitamin D receptor. In this figure, up-arrow shows enhanced activity and down-arrow shows reduced activity.

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
