# Peer review of "Role and Treatment of Insulin Resistance in Patients with Chronic Kidney Disease: A Review"

_nutrients, 2021, doi:10.3390/nu13124349_

Round 1

Reviewer 1 Report

The study presents high quality and deals with important clinical issue, such type of study is needed.  I have only few small remarks that authors should address properly.

There are only some points to correct:

 - please provide the list of abbreviations

 - please add prisma flow diagram

- introduction and discussion section need improvement; please provide information on how your results will translate into clinical practice

- in discussion section please provide study strong points  and study limitation section

- please correct typos

All abovementioned issues are crucial for the credibility of the results.  All the issues need be addressed.

Author Response

The study presents high quality and deals with important clinical issue, such type of study is needed.  I have only few small remarks that authors should address properly.

There are only some points to correct:

 - please provide the list of abbreviations

Response: Thank you for your kind comments. We have added the list of abbreviations at the end of the manuscript.

 - please add prisma flow diagram

Response: We appreciated your helpful comments. We think this manuscript is a clinical review for medical staff and did not summarize new data. Since this manuscript did not include meta-analysis data, the Prisma flow diagram is unsuitable. Next time, we will plan a meta-analysis on the association between vitamin D and IR, where the Prisma flow diagram will be used. Thanks for the meaningful suggestion.

- introduction and discussion section need improvement; please provide information on how your results will translate into clinical practice

Response: We have revised the introduction and discussion sections. This manuscript emphasizes the importance of insulin resistance and discusses the relationship between insulin resistance CKD-MBD, such as vitamin D, phosphorus, and FGF23.

We added the following sentences.

[Introduction]

In addition, a recent study reported that CKD-mineral bone disorder (MBD), phosphorus, and fibroblast growth factor-23 (FGF-23) also affect IR [5]. There is the possibility that IR is improved by CKD-MBD, such as phosphate binder and vitamin D. Vitamin D has also received a great deal of attention in recent years in IR. Clinical studies have been conducted to improve IR with activated vitamin D analog and vitamin D supplementation.

[discussion]

On the other hand, there are opposite results for vitamin D regarding IR[44,76,77]. Several meta-analyses of vitamin D supplementation for IR have been reported. He et al. reported that although vitamin D supplementation had no significant effect on IR, insulin resistance was improved for those with 25(OH)D ≥30 ng/ml by stratified analysis [78]. Other studies also suggested vitamin D supplementation improves IR in prediabetics and polycystic ovary syndrome [79,80]. The evidence of activated vitamin D analog and vitamin D supplementation for IR was insufficient, and well-designed clinical studies are needed.

- in discussion section please provide study strong points  and study limitation section

Response: In the discussion section, we added information about earlier studies and emphasized the strong point of our results. We have added the following to the limitation section, 

As earlier studies reported, calcium-containing phosphate and non-calcium-containing phosphate binders decrease FGF23 [84-86]. Future studies that compare phosphate binder and placebo for IR are needed. In addition, researchers should investigate the differences between non-calcium-containing phosphate and calcium-containing phosphate binders for IR.

This review article has some limitations. First, the clinical evidence of IR is lacking, especially randomized controlled trials that explored the possibility of decreasing IR. Therefore, future well-designed studies will be needed. Second, as IR is regarded as a risk factor for CVD events,  few data exist on other clinical important events, such as fracture and malignancy.

- please correct typos

Response: We are sorry for the typos. We have engaged a professional English language editing service to proofread the manuscript again.

Reviewer 2 Report

This brief review describes the mechanisms that link CKD to IR and how this condition can lead to an increased risk of CVD.

The topic is very interesting as well as hot.

However, this reviewer raises some issues that need to be addressed by authors.

1- At least one other figure or table should be included in the review that more fully shows the mechanisms described by the authors.

2- The review is quite short. However, a topic that is so important and on which numerous articles have been written recently deserves more references:

  • The clinical management of DKD, a model of CKD with high IR, must necessarily include a multifactorial medical approach as has been brilliantly demonstrated recently by the multicenter randomized NID study (Cardiovasc Diabetol (2021) 20:145. doi: 10.1186/s12933-021-01343-1) in which mortality and total MACEs were significantly reduced in few years, with a long durability of this effect. This important issue and above reference should be commented in the text.      
  • It is well known that IR is the strongest pathophysiological link between NAFLD and Metabolic Syndrome. Recent studies have shown that the reduction of IR through the pharmacological eradication of HCV by direct-acting antivirals leads to both a reduction in the onset of type 2 diabetes (Diabetes, Obesity and Metabolism2020, 22(12):2408–2416. doi: 10.1111/dom.14168) and clinical expressions of atherosclerosis (Atherosclerosis2020, 296:40–47. doi: 10.1016/j.atherosclerosis.2020.01.010 - Nutrition, Metabolism & Cardiovascular Diseases (2021) 31, 2345e2353. doi: 10.1016/j.numecd.2021.04.016). These interesting issues as well as the above references deserve to be commented in discussion by the authors.

3- A linguistic review of the manuscript by a native English speaker is required.

Author Response

This brief review describes the mechanisms that link CKD to IR and how this condition can lead to an increased risk of CVD.

The topic is very interesting as well as hot.

However, this reviewer raises some issues that need to be addressed by authors.

1- At least one other figure or table should be included in the review that more fully shows the mechanisms described by the authors.

Response: We appreciate your suggestion. Because this paper focuses on Vitamin D and CKD-MBD for insulin resistance, we added the figure describing the relationship between insulin resistance and vitamin D and CKD-MBD.

2- The review is quite short. However, a topic that is so important and on which numerous articles have been written recently deserves more references:

  • The clinical management of DKD, a model of CKD with high IR, must necessarily include a multifactorial medical approach as has been brilliantly demonstrated recently by the multicenter randomized NID study (Cardiovasc Diabetol (2021) 20:145. doi: 10.1186/s12933-021-01343-1) in which mortality and total MACEs were significantly reduced in few years, with a long durability of this effect. This important issue and above reference should be commented in the text.      
  • It is well known that IR is the strongest pathophysiological link between NAFLD and Metabolic Syndrome. Recent studies have shown that the reduction of IR through the pharmacological eradication of HCV by direct-acting antivirals leads to both a reduction in the onset of type 2 diabetes (Diabetes, Obesity and Metabolism, 2020, 22(12):2408–2416. doi: 10.1111/dom.14168) and clinical expressions of atherosclerosis (Atherosclerosis, 2020, 296:40–47. doi: 10.1016/j.atherosclerosis.2020.01.010 - Nutrition, Metabolism & Cardiovascular Diseases (2021) 31, 2345e2353. doi: 10.1016/j.numecd.2021.04.016). These interesting issues as well as the above references deserve to be commented in discussion by the authors.

Response: Thank you for your very informative comments. We appreciate you for introducing important papers. We have added the four suggested papers to the revised manuscript (Cardiovasc Diabetol (2021) 20:145. doi: 10.1186/s12933-021-01343-1, Diabetes, Obesity and Metabolism, 2020, 22(12):2408–2416. doi: 10.1111/dom.14168, Atherosclerosis, 2020, 296:40–47., Nutrition, Metabolism & Cardiovascular Diseases (2021) 31, 2345e2353) in our manuscript.

A Recent interventional study showed that multifactorial treatment, not only glycol-metabolic control but also hypertension and dyslipidemia by non-pharmacological and pharmacological approaches, prevent CVD events in diabetic CKD patients. This study suggests the importance of multifactorial approach in treating CKD patients[30].

A recent clinical study showed that hepatitis C virus clearance by direct-acting antivirals improved IR [18,19]. In addition, another study reported that hepatitis C treatments reduce CVD events in the prediabetic population [20]. These results suggest the importance of hepatitis C virus for glycemic conditions and the possibility as a future therapeutic target for diabetes mellitus.

3- A linguistic review of the manuscript by a native English speaker is required.

Response: We have engaged a professional English language editing service to proofread the manuscript again.

Round 2

Reviewer 2 Report

No further comments.